# Parental Knowledge about Meningococcal Disease and Vaccination Uptake among 0–5 years Old Polish Children

**DOI:** 10.3390/ijerph16020265

**Published:** 2019-01-18

**Authors:** Marzena Drozd-Dąbrowska, Katarzyna Topczewska, Marcin Korzeń, Anna Sałacka, Maria Ganczak

**Affiliations:** 1Department of Epidemiology and Management, Pomeranian Medical University, Szczecin 71-210, Poland; marzena.drozd@pum.edu.pl (M.D.-D.); katarzyna.topczewska@pum.edu.pl (K.T.); 2Department of Methods of Artificial Intelligence & Applied Mathematics, Westpomeranian University of Technology, Szczecin 71-210, Poland; mkorzen@wi.zut.edu.pl; 3Department of Family Medicine, Pomeranian Medical University, Szczecin 70-204, Poland; anna.salacka@pum.edu.pl

**Keywords:** meningococcal infection, children, parents, knowledge, vaccination, uptake

## Abstract

In Poland, in addition to mandatory, free of charge vaccines, listed in the national immunization schedule, numerous self-paid vaccinations are recommended, including meningococcal vaccination (MV). To assess MV uptake among ≤5-year-old children and to evaluate parental knowledge and attitudes regarding invasive meningococcal disease (IMD). A cross-sectional study was conducted in 2018 among 350 parents (mean age: 32.3 years), attending three randomly selected primary care clinics in Szczecin region, Poland. Anonymous questionnaires were administered to the parents by researchers, present at the time the survey was conducted, to self-complete on a voluntary basis. Chi-square or Fisher’s exact for categorical and Mann–Whitney U test for continuous variables. Variables significantly (*p* < 0.05) associated with ‘good knowledge’ in the bivariate analysis were used to build a logistic regression model. It was found that the response rate was 93.4%, and MV uptake among children was 29.5%. The main knowledge sources were medical staff and media; 72.5% had ever received information about IMD. Only 18.8% of parents self-assessed their knowledge regarding IMD as good; 61.8% scored >50% in the knowledge test 58.9% knew the mode of transmission, 58.7% recognized the severity of meningitis, and 79.7% knew that bacterial meningitis is a vaccine-preventive disease. Knowledge regarding IMD was higher among parents with higher educational level (OR = 3.37; *p* = 0.01), from urban facilities (OR = 2.20; *p* = 0.02), who received previous information about IMD (OR = 2.85; *p* = 0.01) and self-assessed their knowledge as good (OR = 2.59; *p* = 0.04). Low MV coverage among children up to five years old and knowledge gaps about IMD call for awareness campaigns which may increase the coverage. Although educational interventions should cover all parents, those from provincial facilities, representing lower education level need special attention.

## 1. Introduction

Invasive meningococcal disease (IMD), caused by the bacterium *Neisseria meningitidis,* is a public health threat and continues to cause substantial mortality and morbidity [1,2,3]. Based on the polysaccharide capsule, the main virulence factor, thirteen serogroups can be classified of which six (A, B, C, W, X, and Y) are the dominant disease-associated serogroups. IMD may present as meningitis, septicemia, or both, and may cause serious physical disability in up to 20% of all survivors or can be fatal in about 10% of treated cases [2,3]. 

More than three thousand confirmed cases of IMD were reported in EU/EEA countries in 2016 [1]. The overall notification rate was 0.6/100,000 population and country-specific rates ranged from 0.1 to 2.4 [1,4,5]. Poland has a relatively low incidence of IMD among EU countries, with age standardized rates of 0.6/100,000 in 2017 [6]. Currently, every year in Poland, around 200 individuals are diagnosed with this disease [6]. 

In 2016, the majority of confirmed cases in the European region belonged to serogroup B (54%), followed by C (16%) and W (15%), and were notified in infants and young children (8.5/100,000 population in children under one year of age and 2.7 in 1–4-year-olds respectively) with the highest rates reported in Lithuania (41.3) and Ireland (27.2), followed by the UK (11.7) and Poland (11.6) [1]. In Poland serogroup B was a causal agent of about 67.6% of confirmed cases of IMD, while serogroup C of about 26.6%, W–4.4%, Y–1.3%. Most of the infections (81.0%) caused by serogroup B were diagnosed among children up to one year old [7].

In response to a steep increase of IMD cases, different meningococcal conjugate vaccines (MCVs) were systematically introduced in EU countries to all children aged 1–18 years since the beginning of the century [3,8,9]. Such vaccines induce systemic polysaccharide specific IgG levels that are important for individual protection against IMD. Besides individual protection, MCVs establish herd protection by reducing meningococcal acquisition and bacterial transmission to the unprotected population [8]. After initial introduction of MenC conjugate vaccines in Europe in 1999, from 2013, the European Medicines Agency has authorized a new vaccine against meningococcus B, a recombinant protein vaccine including outer membrane vesicles against serogroup B (4CMenB). The vaccine has an acceptable short-term safety profile, providing protection against 73–87% of circulating strains, depending on the country [10,11]. In addition, it provides potential cross-protection against IMD caused by other serogroups [12,13]. In recent years 4CMenB has been introduced into the national routine childhood vaccination program in the UK (2015), Ireland (2016), and Italy (2017) [1,14]. In addition, the vaccine is recommended (but not publicly funded) in Austria, the Czech Republic and Saxony in Germany and recommended for high-risk groups in Belgium, France, Luxembourg, Norway, Portugal, and Spain [1]. 

The immunization schedule in Poland for 2019 includes 11 mandatory vaccines: against tuberculosis, hepatitis B, diphtheria, tetanus, pertussis, poliomyelitis, Haemophilus influenzae type b, pneumococci, measles, mumps, and rubella [15]. The implementation of mandatory vaccinations was an important element of the former Soviet Union guided public health system introduced in all associated countries. Refusal to vaccinate results in starting an administrative procedure, which includes a monetary fine. Even after the system was transformed in 1989, mandatory vaccinations were still maintained with wide parental acceptance, assuring a very high vaccine uptake among children [16]. In addition to mandatory vaccines, vaccinations for meningococcal, rotavirus, varicella, and influenza infections are recommended for Polish children; their cost has to be met by parents [15].

Three types of vaccine against N. meningitidis are licensed: monovalent C, monovalent B, and quadrivalent A+C+Y+W135. According to the National Immunization Program for 2019, vaccination with abovementioned MCVs is recommended, but not mandatory, and can be used from two months of age [4,15]. Vaccinations must be purchased at primary health care sites; the cost of a full series is about 350 USD. According to the official report, 46,489 individuals were vaccinated against N. meningitidis in 2015, the majority (91.4%) were children and young people up to 19 years of age. However, this was less by 10.7% than in 2014 (52,036) [4]. This progressive and alarming reduction of the vaccination uptake, involving this and other mandatory and recommended vaccines, has been observed in Poland in the last few years [17]. 

The decision to immunize a child against N. meningitidis is affected by parental knowledge of the IMD, as well as their knowledge of the vaccine, and attitudes about its use [2,18]. Although numerous studies in different countries have investigated the *N. meningitidis* vaccine uptake, as well as knowledge of IMD and the acceptability of the vaccine among parents [2,19,20,21,22,23], surveys related to this issue are rather scant in Central Europe, including Poland. 

Therefore, overarching aim of this study was to reduce the future frequency of meningococcal transmission among children and to effectively prevent IMD incidence and mortality in this group. Two objectives of the study were defined. The first was to assess the meningococcal vaccination coverage among a sample of Polish aged 0–5 years children from selected primary care clinics (PCC). The second objective was to investigate their parents’ awareness and knowledge regarding IMD, as well as determinants influencing good knowledge level. Such evidence could be used to address and tackle the possible barriers to successful uptake of the vaccine and work on better shaping of interventions that would, in turn, increase coverage in this at-risk population [2]. 

## 2. Materials and Methods

### 2.1. Design and Setting

A cross-sectional study was conducted from January to October 2018 in three randomly selected PCCs. According to medical literature the vaccination uptake is poorer outside larger cities [24,25]. Therefore, we aimed to select PCCs serving patients in urban area (the city of Szczecin with 404,900 inhabitants [25], located in north western Poland) and compare them with practices in less populated urban areas, to ensure representation of different levels of service. Two practices were selected in Szczecin and one in the city of Gryfino (the capital of a neighboring county, with 21,500 inhabitants [26]) with the use of a random-number table.

### 2.2. Study Population and Sampling

The finite population of children 0–5 years old, living in Szczecin and Gryfino for the day 31 December 2017 was 23,751. According to the results of our previous study [27], meningococcal vaccination uptake among Polish children 0–5 years old was 13.3%. With a confidence level (CI) of 95%, and the arbitral relative precision of 6% points on each side, a sample of 264 parents of children 0–5 years old was needed for the purpose of this study [28]. Three hundred and fifty patients were invited to participate. Therefore, the required condition for a minimal sample size was fulfilled. 

All consecutive parents who were present at a selected PCCs with their 0–5 years old child at the time of this study, and who gave informed consent, were invited to participate. They were also informed that they had full rights to withdraw from the survey at any time and that participation would be voluntary. 

### 2.3. Study Instrument and Data Collection

A structured anonymous self-administered questionnaire was used as the main data collection instrument. It was designed by the authors with the use of literature review [2,29]. To increase response rate [30,31], parents were assured of confidentiality of the survey. In addition, questions of a sensitive nature were omitted. Questionnaires (completed by respondents) were administered to the parents by three researchers (M.D-D., K.T., and A.S.), present at a dedicated PCC at the time the survey was conducted. In case of any queries, parents were provided with relevant explanation, as well as any further information that would allow them to correctly understand and properly answer the survey questions. The researchers were trained beforehand about the level of help that could be provided regarding the fulfillment of study questionnaires.

The survey instrument was pilot-tested on 30 parents from one PCC. The respondents’ comments mainly referred to writing a more specific clarification concerning the relatively small number of questions or to adding one more answer or an open question. Therefore, after reviewing the comments and making a number of amendments to improve the clarity of the questions to the study population, the results from the pilot study were included to the main survey. The visits to the PCCs made by the children were accompanied mostly by one of the parents. In the far less common situations when both parents attended, one parent was randomly selected and then asked to fill out the questionnaire. 

The questionnaire queried parents on (1) socio-demographic data; (2) sources of information about IMD; (3) general attitudes towards immunization; (4) awareness and knowledge about IMD; (5) concerns about meningococcal vaccination.

Knowledge was assessed by correctly answering five questions about meningococci as etiological factors of IMD, transmission routes, symptoms/outcomes of IMD, and vaccination. A scale was created to measure knowledge level (from 0 to 5 points). A total knowledge score was then calculated for each respondent. Knowledge scores were set as follows: poor, <3 points (less than 60% correct answers); adequate, 3–5 (≥60%).

The study was approved by the Pomeranian Medical University Ethic Committee (KB-0012/120/02/18).

### 2.4. Statistical Analysis

Categorical data were presented as frequencies with percentages and continuous data as medians. Our primary outcome variable was knowledge regarding meningococcal disease and we aimed to identify variables associated with this outcome. The variables of interest were as follows: age (≤32 years/>32 years), gender (male/female), education level (elementary–vocational/high school-university), marital status (single/married–cohabitate), employment status (employed/unemployed), number of children in the family (≤2/>2), facility location (urban/provincial), receiving information about meningococci (yes/no), self-assessment of knowledge as good/poor-lack of knowledge and concerns about the safety of vaccine against meningococci (yes/no), associated with an outcome variable. Chi-square or Fisher’s exact for categorical and Mann–Whitney U test for continuous variables were used. P values <0.05 were considered statistically significant.

For categorical (binary) variables, as described above, groups were compared using the chi square and Fisher tests. To build a logistic regression model the set of predictors was used with the help of the R MASS package [32,33]. Final associations between predictors and the outcome adjusted for covariates were measured with the use of coefficients of a logistic regression model. Coefficients for binary variables are equal to the natural logarithm of the odds ratio. STATISTI-CA PL, Version 12.5 (StatSoft, Kraków, Poland) and R software (R Foundation for Statistical Computing, Vienna, Austria) were used for data analyses [34].

## 3. Results

### 3.1. Sociodemographic Characteristics

Initially 350 parents were invited, 327 agreed to participate in the study (the response rate 93.4%). The sociodemographic characteristics of parents are presented in Table 1. The mean age of the 327 participants was 32.3 years (range: 17–57 years; SD = 6). Therefore, we arbitrarily decided to choose 32 years of age as the cut off. The majority were females, had graduated high school or university, were married or lived in partnership and had up to two children; almost three-fourths were currently working and lived in urban areas, 85.7% owned a car.

### 3.2. Sources of Knowledge about Meningococcal Disease

The most common sources of knowledge about IMD were medical staff–a family doctor (36.4%), a pediatrician (36.2%), a nurse (13.8%), followed by internet (53.6%) and television (49.4%); this was a multiple-choice question.

### 3.3. Knowledge about Meningococcal Disease

The parental knowledge regarding IMD is presented in Table 2. About two-thirds knew the etiological agent of IMD, 58.9% correctly recognized mode of transmission (through respiratory droplets), 58.7% recognized the severity of meningitis, 79.7% knew that bacterial meningitis could be prevented by vaccination, and 47.0% knew that vaccination against meningococci also protects against sepsis. The mean parental knowledge score was 2.97 (SD = 1.5); 17 of 296 who responded to knowledge questions (5.7%) scored 0 points, 41 (13.8%)—scored 1 point, 55 (18.6%) scored 2 points, 63 (21.3%) scored 3 points, 60 (20.3%)—scored 4 points and 60 (20.3%) scored 5 points. About two-thirds (61.9%) of parents scored >50%. 

Only 60 (18.8%) of parents self-assessed their knowledge as good, 163 (51.1%) as poor, 96 (30.1%) stated they lack adequate knowledge regarding meningococcal disease. In the group which self-assessed their knowledge as good the percentage of respondents who scored >2 was significantly higher when compared to the group which self-assessed their knowledge as poor and lack of knowledge—53/60 (88.3%) versus 140/259 (54.1%); *p* < 0.001 (Table 3). 

### 3.4. Previous Information about Meningococcal Disease

From 327 parents, 309 answered this question. About one quarter of the parents (85; 27.5%) had never received information about IMD. Among those who had received information about IMD, 163/224 (72.8%) had adequate knowledge level versus 23/85 (27.1%) in the group which had not (*p* < 0.001). 

### 3.5. Parental Attitudes Towards Vaccines 

Regarding parental attitudes towards vaccines, 309 (95.3%) stated that vaccines are effective in disease prevention; 285 (89.6%) stated that vaccinating a child is important for others society members; and 180 (56.3%) were concerned about possible side effects of meningococcal vaccination. 

### 3.6. Factors Influencing Knowledge about Meningococcal Infection

The main factors influencing the knowledge about meningococcal disease are presented in Table 3. Parents who were married/cohabitating (*p* = 0.007), with higher education level (*p* < 0.001), having more than two children (*p* = 0.04), owning a car (*p* < 0.001) an those living in urban area (*p* = 0.002) presented the higher knowledge level. There was no statistically significant difference in the knowledge level by parental age, gender, employment status, type of facility, and concerns about vaccine safety.

Regression analyses were performed to assess factors associated with parental knowledge about meningococcal infection (Table 4). Knowledge level was higher among parents with high school/university degree (OR = 3.37; *p* = 0.01), from urban facilities (OR = 2.20; *p* = 0.02), who had ever received information about IMD (OR = 2.85; *p* = 0.01), and those who assessed their knowledge as good (OR = 2.59; *p* = 0.04).

### 3.7. Vaccination Uptake among Children

The vaccination uptake regarding meningococcal vaccine among surveyed children was 95/322 (29.5%).

## 4. Discussion

### 4.1. Results Overview

Our findings indicate that about three quarters of parents had ever received information about IMD. However, only 18.8% of respondents estimated their knowledge regarding this issue as good; the main knowledge sources were medical staff, internet, and television. Overall, parental knowledge on meningococcal disease was mediocre; 61.9% scored >50%. The logistic regression revealed that knowledge level was higher among those with high school/university degree (*p* = 0.01), from facilities located in in urban area (*p* = 0.02), who had ever received information about IMD (*p* = 0.01), and those who self-assessed their knowledge as good (*p* = 0.04). Attitudes toward vaccines were positive in the majority of parents; however, 56.3% were concerned about the safety of the meningococcal vaccine. The meningococcal vaccination uptake among children was 29.5%.

### 4.2. Meningococcal Vaccination Uptake

In previous studies conducted in Poland between 2012–2015 among children from different GP practices, the vaccination uptake was only about 5.0–13.3% [27,35,36], which was a lower uptake than found in this survey. The fact that meningococcal vaccine coverage in such a high-risk group as young children remains low in Poland, when compared to other countries [2,37], is worrying. As an example, 47.3% of Italian parents surveyed by Morone et al. had immunized their children with a MenC vaccine [2]. A possible explanation for the poor vaccination rates among young children in Poland could be the lack of meningococcal vaccine provision within public health insurance. Malerba et al. conducted a European systematic literature review about meningococcal vaccination determinants and found that payment for vaccination is a major barrier [38]. Although there are different vaccination schedules regarding meningococcal vaccination, with different serogroups covered by immunization programs [39], numerous EU member state have included vaccines in their routine immunization programs. These include polysaccharide and conjugate, monovalent and polyvalent vaccine against serogroups A, C, W, and/or Y, and outer membrane vesicle (OMV) vaccines against serogroup B. The specific vaccine use in each country depends on the predominant serogroups, cost and availability [40]. 

Due to a high endemic rate of IMD (11.6 cases/100,000 population per year) in infants and young children [1], Poland should follow examples from the other EU countries and use an appropriate meningococcal vaccine for prevention and/or outbreak response [40]. The highest IMD incidence rate in children <1 year suggests that the most effective prevention strategy is to focus the immunization intervention on this age group. The impact of this vaccine has been documented in several countries with reliable surveillance systems, and includes a direct decrease in incidence rates, as well as indirect benefits due to the induction of herd protection [41].

More than a half of the parents surveyed in this study expressed concerns regarding vaccine safety. This may be partly explained by the fact that our study population received information, not only from medical staff, but also from the internet and television. The latter sources tend to report predominantly on the speculated negative outcomes of vaccines. This result is consistent with some other studies [2,3,18,20,21] which reported meningococcal vaccine side effects as the commonly stated reason for not having a child receive the vaccine. The safety of different types of the vaccine has been documented [42,43], therefore, improving communication tools—i.e., provision of adequate information by physicians or public health authorities—could raise parental awareness regarding this subject. 

Another explanation of the poor meningococcal vaccination coverage among Polish children could be the fact that parental awareness and knowledge related to IMD is inadequate.

### 4.3. Awareness of IMD

Parental awareness of IMD is crucial as it may influence the decision to vaccinate children [44,45]. However, only 72.5% of parents in the present study had ever received information about IMD. This result is much lower than results reported in a study conducted in Italy, where almost all of the parents (95.8%) had heard about meningitis [2]. Though, the Italian study, where higher levels of awareness was observed, was carried out after an Italian Immunization Prevention Plan 2017–2019 introduced the vaccine against meningococcus B in the vaccination schedule for infants.

### 4.4. Knowledge about Meningococcal Infection

This report presents one of the few studies undertaken to evaluate the determinants of knowledge level regarding IMD among parents of children up to five years old. Our results show average parental knowledge, only 62% scored more than 50%. This is worrisome, as adequate knowledge on the subject is vitally important and creates a good starting point for planning future public health campaigns to raise vaccination coverage rates against meningococcal infection to satisfactory levels. 

About two-thirds of our respondents agreed that IMD is a bacterial infection, less than in a study conducted among the South Australian Community (90%). However, the Australian study also included adolescents, and adults, who were non-parents [19]. More than a half of parents in this study knew the mode of meningococci transmission (through respiratory droplets), 80% knew that the vaccine is a preventive measure regarding bacterial meningitis; similar percentages of correct answers were also observed by others [2]. Nevertheless, only 59% of Polish parents recognized the severity of meningitis which might negatively influence their willingness to vaccinate a child against meningococcal disease. In other studies conducted in Italy and Canada, almost all parents agreed that meningitis is a serious disease and the majority of respondents perceived their children as at risk of contracting the disease [2,19]. 

### 4.5. Parental Attitudes Towards Vaccination

Parental attitudes towards vaccinations were positive; 95% of respondents agreed that vaccines are effective in disease prevention and 90% agreed that vaccinating a child is important for other society members. Similar attitudes were also observed in the other Polish study conducted among parents of children attending one GP clinic [35]. However, both results are in contrast with those from the report of Larson et al. on the state of vaccine confidence in the European Union in 2018 which found that confidence has decreased in Poland in the last few years and the country is the least confident towards the importance of vaccines for children [29]. Therefore, a systematic monitoring of parental attitudes towards meningococcal vaccination is needed to detect any possible changes in vaccine confidence.

### 4.6. Determinants of Parental Knowledge about Meningococcal Infection

Important sources of information about meningococcal infection for parents in this study were medical staff, mainly a family doctor, or pediatrician, internet and television which is in line with other studies [2,20,46]. Poor knowledge regarding meningococcal infection observed in our study population may be associated with a number of factors. A multivariable regression analysis revealed that better educated parents had three times higher odds of having good knowledge about meningococcal infection. Results from previous studies also showed that having low educational attainment was more likely to result in lower overall knowledge of IMD [2,47]. It might be hypothesized that parents with a better education background may have more access to health information and may be also more interested in searching for health-related issues, including vaccination. The incidence of IMD is higher in those with lower socioeconomic status and those poorly educated [48], therefore, this result underscores the necessity to eliminate social inequalities in health.

Parents from one specific GP clinic located in a small city were two times less likely to have good knowledge about meningococcal infection. This might be influenced by different education strategies experienced by parents which were used by medical staff with regard to IMD. Such an association due to organizational differences between different facilities and health-promotion related knowledge was also observed previously [24,49,50]. However, Wang et al. found that living in a metropolitan area was more likely to result in lower overall knowledge of IMD [47]. Future research is needed to better assess this issue. 

Additionally, we found that receiving information about meningococcal infection was associated with a three times higher chance of having good knowledge regarding this issue. 

Finally, those who perceived self-knowledge regarding meningococcal infection as high had a two-and-a-half times higher chance of having good knowledge than compared to those who self-assessed their knowledge level as low. This finding might be of importance regarding education and management [2,20]. 

Other authors found age, low household income, low/medium socio-economic status were more likely to have lower overall knowledge of IMD [2,47].

## 5. Limitations

A number of limitations exist in this study. Firstly, it was conducted in three primary care clinics from the West Pomeranian region; therefore, the findings do not necessarily apply to parents from other Polish regions. Therefore, further studies at a national level would be of value. Secondly, parents filled out the questionnaires in the presence of the research team members, and thus may have given more sociably desirable responses [19], this might also introduce a response bias in the study. However, the fact that the questionnaires were completed anonymously should result in reducing this bias. Finally, the cross-sectional study which was used in this survey is less powerful for evaluating risk factors than other analytical study designs [51]. 

## 6. Implications for Educational Interventions 

The results of the study showed that being informed previously about IMD was associated with for times greater odds of having good knowledge regarding this topic. However, healthcare providers are not yet fully providing information about IMD; GPs and pediatricians were sources of IMD knowledge for only about 30% of surveyed parents. Therefore, first-line health care workers should increase their role in information delivery regarding this important topic. Such professional sources would guarantee clear, unbiased information about IMD. Education should be initiated during pregnancy and at delivery wards, with the help of neonatal staff, then expanded during home visits soon after delivery, and continued at the visits made by parents of infants and toddlers to the PCCs [27].

In addition, national media campaigns oriented to educate parents on IMD and motivating them to vaccinate their children against meningococci could also be used as a supportive tool.

Future research, presumably at a national level, should focus on assessing parental preferences regarding IMD information delivery channels and on their educational needs, as well as potential barriers for meningococcal vaccination and factors influencing vaccine hesitancy. 

## 7. Conclusions

The relatively high incidence of IMD among Polish infants and young children [1] and low meningococcal vaccine uptake observed in this study leads to a call for the serious consideration that this vaccine become mandatory in Poland. Governmental support that would make the vaccination free of charge, or at least partly funded, could positively influence uptake.

Poor parental awareness about the IMD and knowledge gaps necessitate national information campaigns, as well as continuous guidance and support from health care workers at the community level, which may increase awareness and knowledge and improve vaccination coverage among children. Although educational interventions should cover all parents, those from provincial facilities, representing a lower education level, need special attention.

## Figures and Tables

**Table 1 ijerph-16-00265-t001:** Characteristics of parents of children up to five years old; Szczecin region, Poland; 2018, *n* = 327.

Variable	*n*	%
Age		
≤32 years	166	50.8
>32 years	161	49.2
Gender of the parent		
Male	46	14.1
Female	281	85.9
Place of residence		
Rural area	87	26.6
Urban area	240	73.4
Facility location		
Provincial (Gryfino)	150	45.9
Urban (Szczecin)	177	54.1
Education level		
Primary/vocational	61	18.7
High school/university	266	81.3
Marital status		
Married/Cohabitating	271	82.9
Single	56	17.1
Number of children		
≤2	282	86.2
>2	45	13.8
Employment status		
Working	243	74.3
Not working	84	25.7
Owning a car		
Yes	276	85.7
No	46	14.3

**Table 2 ijerph-16-00265-t002:** Parental knowledge about IMD; Poland, 2018, *n* = 327

Statement	Correct Answer	True		False	
		*n*	%	*n*	%
IMD is caused by bacteria	yes	222	67.9	105	32.1
IMD is a life-threatening disease	yes	183	58.7	129	41.3
Bacterial meningitis is transmitted by droplets	yes	179	58.9	125	41.1
Bacterial meningitis could be prevented by vaccinations	yes	243	79.7	62	20.3
Vaccination against meningococci also protects against sepsis	yes	147	47.0	166	53.0

**Table 3 ijerph-16-00265-t003:** Factors influencing the knowledge regarding meningococci by bivariate analysis; Szczecin region, Poland; 2018, *n* = 327.

Variable	Adequate Knowledge >2		Poor Knowledge 0–2		*p*
	*n*	%	*n*	%	
Sociodemographic variables					
Age					
≤32 years	100	30.6	66	20.2	0.91
>32 years	98	29.9	63	19.3	
Gender					
Male	25	7.6	21	6.4	0.35
Female	173	52.9	108	33.1	
Place of residence					
Rural area	41	12.5	46	14.1	0.002
Urban area	157	48.0	83	25.4	
Education level					
Primary/vocational education	21	6.4	39	11.9	<0.001
Secondary/high education	177	54.2	90	27.5	
Marital status					
Married/Cohabitating	173	52.9	98	30.0	0.007
Single	25	7.6	31	9.5	
Number of children					
≤2	177	54.1	105	32.1	0.04
>2	21	6.4	24	7.4	
Employment status					
Working	153	46.8	90	27.5	0.13
Not working	45	13.8	39	11.9	
Owning a car					
Yes	181	55.3	100	30.6	<0.001
No	17	5.2	29	8.9	
Type of facility					
Urban	107	32.7	70	21.5	0.89
Provincial	91	27.8	59	18.0	
Other variables					
Had ever received information about IMD					
Yes	163	52.8	61	19.7	<0.001
No	23	7.4	62	20.1	
Self-assessed knowledge as good					
Yes	53	16.6	7	2.2	<0.001
No	140	43.9	119	37.3	
Concerned about the safety of MV					
Yes	118	36.8	62	19.4	0.07
No	78	24.4	62	19.4	

**Table 4 ijerph-16-00265-t004:** Logistic regression model: association of the knowledge level about meningococci with selected variables (odds ratio (OR) estimates and 95% confidence intervals (CIs) of OR estimates); Poland, 2018; *n* = 327.

Variable	OR	CI
Parental age: >32	1.03	0.97–1.09
Gender: father	0.52	0.22–1.22
Marital status: married/cohabitating	1.77	0.85–3.67
Residence: urban	1.74	0.84–3.64
Education: high school/university	3.37	1.31–8.85
Employment: yes	1.18	0.56–2.42
Owning a car: yes	1.11	0.40–3.06
Number of children: ≤2	1.84	0.64–5.30
Facility: urban	2.20	1.16–4.33
Had ever received information about IMD	2.85	1.29–6.44
Self–assessed knowledge: good	2.59	1.08–6.83
Source of knowledge: GP	1.64	0.82–3.31
Concerned about the safety of MV: yes	0.99	0.53–1.84

Odds ratio (OR) = ratio between the two categories tested in each variable, controlling for other variable.

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
