# Peer review of "Parental Knowledge about Meningococcal Disease and Vaccination Uptake among 0–5 years Old Polish Children"

_ijerph, 2019, doi:10.3390/ijerph16020265_

Reviewer 1 Report

Marzena Drozd-Dąbrowska et al did a survey by questionnaire to evaluate parental knowledge and attitudes regarding invasive meningococcal disease (IMD) and meningococcal vaccination (MV) among 327 parents with 0-5 years old children in Polish. The author found that most people don’t have sufficient knowledge about IMD and this may be closely associated with lower education level. After carefully reading, I have following concerns.

Major:

The author revealed that “72% of the parents who 196 had ever heard about meningococcal disease showed a high level of knowledge”. As a result, to enhance people’s awareness about IMD, they suggested that “lower education level should be targeted first”. Since 81.3% participants had graduated high school or university, however, while Only 18.8% of parents self-assessed 27 knowledge regarding IMD as good, suggesting that even high educated parents who heard about IMD did not take it seriously.

Also, as most of their knowledge about IMD are from followed by internet (53.6%) and television (49.4%). I think the author may also add a suggestion to offer these parents formal sources to access enough and correct information about this disease such as distribute a specific book among antenatal classes etc.

Moreover, as the vaccination uptake regarding meningococcal vaccine among surveyed children was as low as 95/322(29.5%), this is due to “more than a half of the parents surveyed in this study expressed concerns regarding vaccine safety”. The author also noted that internet and television tend to report predominantly on the speculated negative outcomes of vaccines. This make offer formal access to parents important. 

Then, we are curious how these 95 children distributed? Whether parents have higher education level or those did not realize the safety of vaccine are more willing to immune their children.

Minor:

Some words in figure 1 are overlapped with each other.

Author Response

We would like to thank the Reviewer for the time spent as well as for all valuable comments which allowed us to improve the manuscript.

Please, find bellow our point-by-point response to the Reviewer's comments:

Major:

1.       The author revealed that “72% of the parents who had ever heard about meningococcal disease showed a high level of knowledge”. As a result, to enhance people’s awareness about IMD, they suggested that “lower education level should be targeted first”. Since 81.3% participants had graduated high school or university, however, while Only 18.8% of parents self-assessed knowledge regarding IMD as good, suggesting that even high educated parents who heard about IMD did not take it seriously.

We agree with the Reviewer that all parents of children 0-5 years old should get more information about IMD. However, in Poland health expenditure per capita is one of the lowest in the EU; this has an influence on vaccination policy. Therefore, in our opinion, future interventions should be more tailored, focusing especially, though not only, on vulnerable subgroups identified in this study, which are particularly difficult to reach regarding educational interventions, such as less educated parents from provincial facilities.

According to the Reviewer’s suggestion this has been changed as follows:

Although educational interventions should cover all parents, those from provincial facilities, representing lower education level need special attention.

2.       Also, as most of their knowledge about IMD are from followed by internet (53.6%) and television (49.4%). I think the author may also add a suggestion to offer these parents formal sources to access enough and correct information about this disease such as distribute a specific book among antenatal classes etc. Moreover, as the vaccination uptake regarding meningococcal vaccine among surveyed children was as low as 95/322(29.5%), this is due to “more than a half of the parents surveyed in this study expressed concerns regarding vaccine safety”. The author also noted that internet and television tend to report predominantly on the speculated negative outcomes of vaccines. This make offer formal access to parents important.

We would like to thank the Reviewer for this valuable comment. We have added this to the Discussion section (Implications for educational interventions) as follows:

The results of the study showed that being informed previously about IMD was associated with for times greater odds of having good knowledge regarding this topic. However, healthcare providers are not yet fully providing information about IMD; GPs and paediatricians were sources of IMD knowledge for only about 30% of surveyed parents.  Therefore, first-line health care workers should increase their role in information delivery regarding this important topic. Such professional sources would guarantee clear, unbiased information about IMD. Education should be initiated during pregnancy and at delivery wards, with the help of neonatal staff, then expanded during home visits soon after delivery, and continued at the visits made by parents of infants and toddlers to the PCCs

3.       Then, we are curious how these 95 children distributed? Whether parents have higher education level or those did not realize the safety of vaccine are more willing to immune their children.

We would like to thank the Reviewer for this valuable comment. We have looked at abovementioned potential factors influencing the meningococcal vaccine uptake which are as follows:

a.       Parental education level and child vaccination

Parents with primary/vocational level: 8/57 (2.5%) children vaccinated vs Parents - high school/university graduates: 87/265 (27.0%) children vaccinated; p=0.005

b.       Parental concerns about the MV safety and child vaccination

Concerned Parents: 44/177 (24.8%) children vaccinated vs Parents with no concerns: 49/139 (35.2%) children vaccinated; p=0.44

4.       Minor: Some words in figure 1 are overlapped with each other

Figure 1 has been checked according to the Reviewer’s suggestion.

Reviewer 2 Report

I enjoyed reading your work, but have some suggestions as to how it could be further enhanced:

Abstract

Are vaccines mandatory? In the UK, we have an immunisation schedule whereby people are recommended (and encouraged to get) particular ‘routine’ vaccinations as part of the national programme, but I don’t think the vaccines are mandatory, per se.

I think you need to expand on the methodology and cut back on the results (for example, there is no mention of data analysis or that participation was voluntary/parents were invited to participate).

‘the IMD’ should probably just be ‘IMD’.

I think the following statement needs rephrased or further explained: “Only 18.8% of parents self-assessed knowledge regarding IMD as good, however, 61.8% scored >50%”.

I think the following statement needs to be written without using a ‘-‘  i.e. “…79.7% - that bacterial meningitis is a vaccine-preventive disease.”  Maybe “…and 79.7% knew that meningitis is a vaccine….

To say that knowledge about IMD was higher in those “who had ever heard about IMD” seems implicit. Maybe it needs rephrased because currently it doesn’t seem like a key finding.

You conclude about the ‘alarmingly low coverage’ but how is this alarming if it’s around 30% and you’ve already stated in the introduction that “meningococcal vaccination (MV) is yet to be introduced as mandatory in Poland.”

Main text

As mentioned before, are some vaccines mandatory (i.e. compulsory) in Poland? In the UK, we have an immunisation schedule whereby people are recommended (and strongly encouraged to get) particular ‘routine’ vaccinations as part of the national programme, but the vaccines are not mandatory/compulsory for children or adults, per se.

It might be better to have an overarching aim and several specific objectives.

Line 85 – is it “the Central Europe” or just “central Europe”?

Lines 96-101 – how many parents did you invite to participate in the study? You state about the city of Szczecin with 404,900 inhabitants and the city of Gryfino with 21,500 inhabitants, and that “the study participants were consecutive parents of 0-5 years old children who attended PCC.”

Line 97 – perhaps you could expand on how you randomly selected the study sites and your recruitment strategy

Line 99 - how much time were parents given between being invited to participate and participation (and it’s not clear how they found out about the study or how they were invited to participate). What happens if it was a carer with the child (like a grandparent), rather than a ‘parent’?

Line 104 – the questionnaire was “distributed by the research team” and in the abstract it was described as an “investigator-administered questionnaire”. What role did the investigators play – did they just distribute to the participants for them to self-complete or were the investigators involved in the actual completion of the questionnaire (and could clarify issues etc.)?

Line 105 – why include the pilot data in the study?

What measures did you employ from the outset to maximise the response rate of your questionnaire?

Line 119 – can you provide a bit more justification about your variables – so, for example, why 32 years of age as the cut off. Is this based on other findings in the literature? Why group high school and university together rather than having secondary education and tertiary education as two separate entities.

Line 135 - Response rate is stated as 93.4% but it isn’t obvious in the method as to the number of people who were invited to participate. Presumably if there were 327 participants, you initially invited 350?

Line 137 – “almost three-fourth” could be changed to “almost three-quarters” (and in other places where you refer to ‘fourths’, these could also potentially be changed to ‘quarters’).

There is some repetition between the text and tables, so you could make the text more succinct/a top-line summary and then signpost the reader to the table for comprehensive information (if word limit is an issue).

Table 2 “Bacterial meningitis could be protected by vaccinations” - do you mean “Bacterial meningitis could be prevented by vaccinations.”

Table 2 “Vaccination against meningococci protects sepsis” – do you mean “Vaccination against meningococci also protects against sepsis”

Line 167 – “Among those who had heard of an infection” – is this ‘heard of any infection” as it is not how you phrased it in the abstract (and see my point about the phrase in the abstract too)?

Table 3 – ‘father’ and ‘mother’ are not really ‘gender’ – consider changing to male and female as per your previous description of the variables.

Table 3  - you shouldn’t have ‘<0.05’ as well as ‘<0.001’ (you have 0.04 and 0.002 as well).

Discussion/conclusion – it would be good to have a summary of your main implications/suggestions for practice and where future research should focus.

Author Response

We would like to thank the Reviewer for the time spent on our manuscript and all valuable comments which helped us to improve the paper.

Our point-to-point response to the Reviewer's comments:

Abstract

1.       Are vaccines mandatory? In the UK, we have an immunisation schedule whereby people are recommended (and encouraged to get) particular ‘routine’ vaccinations as part of the national programme, but I don’t think the vaccines are mandatory, per se.

We would like to thank the Reviewer for this valuable comment. The more detailed explanation of an immunization schedule has been added to the Abstract as follows:

In Poland, in addition to mandatory, free of charge vaccines, listed in the national immunization schedule, numerous self-paid vaccinations are recommended, including meningococcal vaccination (MV).

2.       I think you need to expand on the methodology and cut back on the results (for example, there is no mention of data analysis or that participation was voluntary/parents were invited to participate).

According to the Reviewer’s comment this information has been added to the abstract and to the Methods section of the manuscript as follows:

Abstract

Anonymous questionnaires were administered to the participants by researchers, present at the time the survey was conducted, to self-complete on a voluntary basis. Chi-square or Fisher's exact for categorical and Mann-Whitney’s test for continuous variables were used. To build a logistic regression model, a stepwise backward selection procedure was performed.

Methods

All consecutive parents who were present at selected PCCs with their 0-5 years old child at the time this study was conducted, and who gave informed consent, were invited to participate.  They were also informed that they had full rights to withdraw from the survey at any time and that their participation would be voluntary.

3.        ‘the IMD’ should probably just be ‘IMD’.

Regarding the Reviewer’s comment this has been changed in the manuscript.

4.       I think the following statement needs rephrased or further explained: “Only 18.8% of parents self-assessed knowledge regarding IMD as good, however, 61.8% scored >50%”.

Regarding the Reviewer’s comment this has been changed in the Abstract as follows:

Only 18.8% of parents self-assessed their knowledge regarding IMD as good; 61.8% scored >50% in the knowledge test (58.9% knew the mode of transmission, 58.7% recognized the severity of meningitis, and 79.7% knew that bacterial meningitis is a vaccine-preventive disease).

5.       I think the following statement needs to be written without using a ‘-‘  i.e. “…79.7% - that bacterial meningitis is a vaccine-preventive disease.”  Maybe “…and 79.7% knew that meningitis is a vaccine….

Regarding the Reviewer’s comment this has been changed in the manuscript.

6.       To say that knowledge about IMD was higher in those “who had ever heard about IMD” seems implicit. Maybe it needs rephrased because currently it doesn’t seem like a key finding.

We would like to thank the Reviewer for this valuable comment. We have rephrased this in the  Abstract as follows:

Knowledge regarding IMD was higher among parents (…) who received previous information about IMD (OR=3.82; p<0.001) (…).

7.       You conclude about the ‘alarmingly low coverage’ but how is this alarming if it’s around 30% and you’ve already stated in the introduction that “meningococcal vaccination (MV) is yet to be introduced as mandatory in Poland.”

According to the Reviewer’s comment the adjective “alarmingly” has been deleted.

Main text

8.       As mentioned before, are some vaccines mandatory (i.e. compulsory) in Poland? In the UK, we have an immunisation schedule whereby people are recommended (and strongly encouraged to get) particular ‘routine’ vaccinations as part of the national programme, but the vaccines are not mandatory/compulsory for children or adults, per se.

In Poland vaccinations included in the immunization schedule are mandatory. This means that every child residing in Poland can receive vaccines refunded by the state, but it also means that parents are obliged come to vaccination appointments. Refusal to vaccinate usually means triggering an administrative procedure, which typically involves a monetary fine.

At birth, each child receives an immunisation card which is stored at the general practitioner (GP) office and used to monitor immunization schedule progress. Based on this card, the GP contacts parents for well-baby visits and administers scheduled vaccines as part of developmental monitoring. The current immunization schedule (2019) includes 11 mandatory vaccines, against tuberculosis, hepatitis B, diphtheria, tetanus, pertussis, poliomyelitis, Haemophilus influenzae type b, pneumococci, measles, mumps and rubella. The immunization schedule also includes a separate section describing which vaccines are recommended, but their cost has to be met by parents.

The introduction of mandatory vaccinations was part of the former Soviet Union guided public health system implemented in all countries associated with the Soviet Union. Even after the democratic reforms began in 1989, mandatory vaccinations were maintained and were widely accepted by Poles, assuring a very high vaccine uptake. 

Regarding the Reviewer’s comment a more detailed explanation of the immunization schedule has been added to the Introduction section as follows:

The immunization schedule in Poland for 2019 includes 11 mandatory vaccines: against tuberculosis, hepatitis B, diphtheria, tetanus, pertussis, poliomyelitis, Haemophilus influenzae type b, pneumococci, measles, mumps and rubella [15]. The implementation of mandatory vaccinations was an important element of the former Soviet Union guided public health system introduced in all associated countries. Refusal to vaccinate results in starting an administrative procedure, which includes a monetary fine. Even after the system was trans-formed in 1989, mandatory vaccinations were still maintained with wide parental acceptance, assuring a very high vaccine uptake among children [16]. In addition to mandatory vaccines, vaccinations for meningococcal, rotavirus, varicella and influenza infections are recommended for Polish children; their cost has to be met by parents [15].

9.       It might be better to have an overarching aim and several specific objectives.

According to the Reviewer’s suggestion, this has been corrected as follows:

Therefore, the overarching aim of this study was to reduce the future frequency of meningococcal transmission among children and to effectively prevent IMD incidence and mortality in this group. Two objectives of the study were also defined. The first was to assess meningococcal vaccination coverage among a sample of Polish children, aged 0-5 years from selected primary care clinics. The second objective was to investigate parental awareness and knowledge regarding IMD, as well as determinants influencing good knowledge level. Such evidence could be used to address and tackle the possible barriers to successful uptake of the vaccine and work on better shaping of interventions that would, in turn, increase coverage in this at-risk population [2].

10.    Line 85 – is it “the Central Europe” or just “central Europe”?

Regarding the Reviewer’s comment this has been changed in the Introduction section of the manuscript.

11.    Lines 96-101 – how many parents did you invite to participate in the study? You state about the city of Szczecin with 404,900 inhabitants and the city of Gryfino with 21,500 inhabitants, and that “the study participants were consecutive parents of 0-5 years old children who attended PCC.”

According to the Reviewer’s suggestion we have expanded the sampling description as follows:

According to medical literature the vaccination uptake is poorer outside larger cities [24,25]. Therefore, we aimed to select PCCs serving patients in urban area (the city of Szczecin with 404,900 inhabitants [25], located in north western Poland) and compare them with practices in less populated urban areas, to ensure representation of different levels of service. Two practices were selected in Szczecin and one in the city of Gryfino (the capital of a neighbouring county, with 21,500 inhabitants [26]) with the use of a random-number table.

The finite population of children 0-5 years old, living in Szczecin and Gryfino in the West Pomeranian region, Poland, for the day 31.12.2017 was 23,751. According to the results of our previous study [], meningococcal vaccination uptake among Polish children 0-5 years old was 13.3%. With a confidence level (CI) of 95%, and the arbitral relative  precision of 6% points on each side, a sample of 264 parents of children 0-5 years old was needed for the purpose of this study []. Three hundred and fifty patients were invited to participate. Therefore, the required condition for a minimal sample size was fulfilled.

All consecutive parents who were present at a selected PCCs with their 0-5 years old child at the time of this study, and who gave informed consent, were invited to participate. They were also informed that they had full rights to withdraw from the survey at any time and that participation would be voluntary.

12.    Line 97 – perhaps you could expand on how you randomly selected the study sites and your recruitment strategy

According to medical literature the vaccination uptake is poorer outside larger cities. Therefore, we aimed to select PCCs serving patients in the urban area (the city of Szczecin with 404,900 inhabitants) and compare them with practices in less populated urban areas (equal or less than 50,000 inhabitants), to ensure representation of  different levels of service. To select the latter type of practices multistage sampling was used. Firstly, with the use of a random-number table, 1 county (Gryfino) was selected randomly out of 4 neighbouring counties in the West Pomeranian region, Poland. Then, with the help of a list of PCCs obtained from the local health department 1 practice was selected out of 5 in the county capital. In the case of the city of Szczecin, 2 practices were selected randomly out of 73 PCCs.

As mentioned before, according to the Reviewer’s comment we have expanded on the selection method as follows:

A cross-sectional study was conducted from January to October 2018 in 3 randomly selected PCCs. According to medical literature the vaccination uptake is poorer outside larger cities [24,25]. Therefore, we aimed to select PCCs serving patients in urban area (the city of Szczecin with 404,900 inhabitants [25], located in north western Poland) and compare them with practices in less populated urban areas, to ensure representation of different levels of service. Two practices were selected in Szczecin and one in the city of Gryfino (the capital of a neighbouring county, with 21,500 inhabitants [26]) with the use of a random-number table.

13.    Line 99 - how much time were parents given between being invited to participate and participation (and it’s not clear how they found out about the study or how they were invited to participate). What happens if it was a career with the child (like a grandparent), rather than a ‘parent’?

We would like to thank the Reviewer for this valuable comment. A more detailed explanation of a recruitment procedure has been added to the Methods section as follows:

All consecutive parents who were present at a selected PCCs with their 0-5 years old child at the time of this study, and who gave informed consent, were invited to participate. They were also informed that they had full rights to withdraw from the survey at any time and that participation would be voluntary.

The visits to the PCCs made by the children were accompanied mostly by one of the parents. In the far less common situations when both parents attended, one parent was randomly selected and then asked to fill out the questionnaire. There was no situation where a guardian, rather than a parent attended the PCC with the child, during the study.

14.    Line 104 – the questionnaire was “distributed by the research team” and in the abstract it was described as an “investigator-administered questionnaire”. What role did the investigators play – did they just distribute to the participants for them to self-complete or were the investigators involved in the actual completion of the questionnaire (and could clarify issues etc.)?

Regarding the Reviewer’s comment the relevant information has been added to the Methods section of the manuscript as follows:

Questionnaires were administered to the participants to self-complete by three researchers (M.D-D., K.T. and A.S.) who were present at a dedicated PCC at the time the survey was conducted. In case of any queries, parents were provided with relevant explanation, as well as any further information that would allow them to correctly understand and properly answer the survey questions.

15.    Line 105 – why include the pilot data in the study?

The survey was pilot-tested on 30 parents from one PCC. The respondents’ comments referred mainly to writing a more specific clarification concerning the relatively small number of questions or to adding one more answers or an open question. Therefore, we decided to include the results in the study after reviewing the comments from the pilot study and making a few amendments to improve the clarity of the questions to the study population.

According to the Reviewer’s comment this has been changed in the Methods section of the manuscript as follows:

The survey was pilot-tested on 30 parents from one PCC. The respondents’ comments mainly referred to writing a more specific clarification concerning the relatively small number of questions or to adding one more answer or an open question. Therefore, after reviewing the comments and making a number of amendments to improve the clarity of the questions to the study population, the results from the pilot study were included to the main survey.

16.    What measures did you employ from the outset to maximise the response rate of your questionnaire?

The fact that questionnaires were administered to the participants to self-complete by three researchers (M.D-D., K.T. and A.S.), present at a dedicated PCC at the time the survey was conducted could influence good response rate. In case of any queries, parents were provided with relevant explanation, as well as any further information that would allow them to correctly understand and properly answer the survey questions.

Furthermore, the fact that researchers involved in data collection were health care workers (physicians and a paramedic), rather than outside interviewers, could have also increased the response rate.

17.    Line 119 – can you provide a bit more justification about your variables – so, for example, why 32 years of age as the cut off. Is this based on other findings in the literature?

The mean age of the study population was 32.3 years. Therefore, we arbitrary decided to choose 32 years of age as the cut off.

According to the Reviewer’s comment this has been added to the Methods section of the manuscript as follows:

The mean age of the 327 participants was 32.3 years (range: 17–57 years; SD=6). Therefore, we arbitrary decided to choose 32 years of age as the cut off.

18.    Why group high school and university together rather than having secondary education and tertiary education as two separate entities.

In Poland, graduates of primary education (for 7-15 years old students) or vocational education (for 15-18 years old) are routinely classified as less educated than graduates of high schools and universities. Therefore, we arbitrary combined the first two and the latter two ones to compare their knowledge level regarding IMD.

19.    Line 135 - Response rate is stated as 93.4% but it isn’t obvious in the method as to the number of people who were invited to participate. Presumably if there were 327 participants, you initially invited 350?

This is correct. Initially 350 parents were invited, 327 agreed to participate in the study. According to the Reviewer’s comment this has been added to the Methods section of the manuscript as follows:

Initially 350 parents were invited, 327 agreed to participate in the study (the response rate 93.4%).

20.    Line 137 – “almost three-fourth” could be changed to “almost three-quarters” (and in other places where you refer to ‘fourths’, these could also potentially be changed to ‘quarters’).

According to the Reviewer’s comment this has been changed in the Results section of the manuscript.

21.    There is some repetition between the text and tables, so you could make the text more succinct/a top-line summary and then signpost the reader to the table for comprehensive information (if word limit is an issue).

According to the Reviewer’s comment some detailed information, such as numerators and denominators as well as percentages were deleted from the text to avoid repetition between the text and the tables.

22.    Table 2 “Bacterial meningitis could be protected by vaccinations” - do you mean “Bacterial meningitis could be prevented by vaccinations.”

We are sorry for this language mistake. According to the Reviewer’s comment this has been changed in Table 2.

23.    Table 2 “Vaccination against meningococci protects sepsis” – do you mean “Vaccination against meningococci also protects against sepsis”

Again, we are sorry for this unprecise translation from Polish language. According to the Reviewer’s comment this has been changed in in Table 2.

24.    Line 167 – “Among those who had heard of an infection” – is this ‘heard of any infection” as it is not how you phrased it in the abstract (and see my point about the phrase in the abstract too)?

Again, we would like to thank the Reviewer for this valuable comment. We have rephrased this in the  Results and Discussion section as follows:

“…had never received information about IMD”.

25.    Table 3 – ‘father’ and ‘mother’ are not really ‘gender’ – consider changing to male and female as per your previous description of the variables.

According to the Reviewer’s comment this has been changed to “male” and “female” in the Results section of the manuscript.

26.    Table 3  - you shouldn’t have ‘<0.05’ as well as ‘<0.001’ (you have 0.04 and 0.002 as well).

We are sorry, a mistake was made regarding calculations of one specific value, i.e. “had never received information about IMD”. There should be “<0.001” instead of “<0.05”.

Regarding p values presented in Table 3, we arbitrary decided that values less than 0.000 would be presented as <0.001.

27.    Discussion/conclusion – it would be good to have a summary of your main implications/suggestions for practice and where future research should focus.

According to the Reviewer’s suggestion this has been added at the Discussion section as follows:

The results of the study showed that being informed previously about IMD was associated with for times greater odds of having good knowledge regarding this topic. However, healthcare providers are not yet fully providing information about IMD; GPs and paediatricians were sources of IMD knowledge for only about 30% of surveyed parents.  Therefore, first-line health care workers should increase their role in information delivery regarding this important topic. Such professional sources would guarantee clear, unbiased information about IMD. Education should be initiated during pregnancy and at delivery wards, with the help of neonatal staff, then expanded during home visits soon after delivery, and continued at the visits made by parents of infants and toddlers to the PCCs

In addition, national media campaigns oriented to educate parents on IMD and motivating them to vaccinate their children against meningococci could also be used as a supportive tool.

Future research, presumably at a national level, should focus on assessing parental preferences regarding IMD information delivery channels and on their educational needs, as well as potential barriers for meningococcal vaccination and factors influencing vaccine hesitancy.

Reviewer 3 Report

The study evaluated the Meningococcal vaccination (MV) uptake among ≤5 year old children and to evaluate parental knowledge and attitudes 23 regarding invasive meningococcal disease (IMD). The paper is interesting but some major questions should be addressed:

1. Introduction: Backgrounds and study rationale are clear. Just a minor comment - is there any government support for the MV vaccine? If not, is there any potential for further policy revision? Let say comparing with MMR vaccine.

2. Materials and Methods: Please describe the purpose of the pilot test.

3. The study lacks a sample size (or power) determination.

4. How did the authors treat with the missing data?

5. Results: Table 1: Why cut-off of 32 years old was used? If there is not any specific meaning, I would suggest keep using "mean age 32.3 years, range: 17–57 years; SD=6" mentioned in the main text.

6. Why "Owing a car" was a variable of interest in the study? Please explain.

7. Why stepwise selection was used? It seems the authors did not intent to build a predction model for that. If the objective is to assess the independent association with the knowledge, all variables should include in the model with doing ANY variable selection procedure.

8. I have no idea why a desicion tree should be presented. No additional information was drawn to the paper.

Author Response

We would like to thank the Reviewer for the time spent on our manuscript and all valuable comments which helped us to improve the paper.

Our point-to -point response to the Reviewer's comments:

1.       Introduction: Backgrounds and study rationale are clear. Just a minor comment - is there any government support for the MV vaccine? If not, is there any potential for further policy revision? Let say comparing with MMR vaccine.

We would like to thank the Reviewer for this valuable comment.

In Poland vaccinations included in the immunization schedule are mandatory. This means that every child residing in Poland can receive vaccines refunded by the state, but it also means that parents are obliged to show up on vaccination visits. Refusal to vaccinate usually means triggering an administrative procedure, which typically involves a monetary fine.

Each child at birth receives an immunisation card which is stored at the general practitioner (GP) office and used to monitor the immunization schedule progress. Based on this card, the GP is calling parents for well-baby visits and administers scheduled vaccines as part of developmental monitoring. The current immunization schedule (2019) includes 11 mandatory vaccines, against tuberculosis, hepatitis B, diphtheria, tetanus, pertussis, poliomyelitis, Haemophilus influenzae type b, pneumococci, measles, mumps and rubella. The immunization schedule includes also a separate section describing which vaccines are recommended, but their cost has to be paid by parents.

The introduction of mandatory vaccinations was part of the former Soviet Union guided public health system implemented in all countries associated with the Soviet Union. Even after the democratic reforms starting in 1989, the mandatory vaccinations were maintained and were widely accepted by Poles, assuring a very high vaccine uptake. 

Regarding the Reviewer’s comment the more detailed explanation of an immunization schedule has been added to the Introduction section as follows:

The immunization schedule in Poland for 2019 includes 11 mandatory vaccines: against tuberculosis, hepatitis B, diphtheria, tetanus, pertussis, poliomyelitis, Haemophilus influenzae type b, pneumococci, measles, mumps and rubella [15]. The implementation of mandatory vaccinations was an important element of the former Soviet Union guided public health system introduced in all associated countries. Refusal to vaccinate results in starting an administrative procedure, which includes a monetary fine. Even after the system was transformed in 1989, mandatory vaccinations were still maintained with wide parental acceptance, assuring a very high vaccine uptake among children [16]. In addition to mandatory vaccines, vaccinations for meningococcal, rotavirus, varicella and influenza infections are recommended for Polish children; their cost has to be met by parents [15].

2. Materials and Methods: Please describe the purpose of the pilot test.

The survey was pilot-tested on 30 parents from one PCC in terms to assess the clarity of the questions and make possible amendments. Respondents’ comments referred mainly to writing a more specific clarification concerning relatively small number of questions or to adding one more answer/an open question. Therefore, we decided to include the results in the study after reviewing the comments from the pilot study and making a few amendments to improve the clarity of the questions to the study population.

According to the Reviewer’s comment this has been changed in the Methods section of the manuscript as follows:

The survey instrument was pilot-tested on 30 parents from one PCC. The respondents’ comments mainly referred to writing a more specific clarification concerning the relatively small number of questions or to adding one more answer or an open question. Therefore, after reviewing the comments and making a number of amendments to improve the clarity of the questions to the study population, the results from the pilot study were included to the main survey.

3. The study lacks a sample size (or power) determination.

We would like to thank the Reviewer for this valuable comment.

The finite population of children 0-5 years old, living in Szczecin and Gryfino in the West Pomeranian region, Poland, for the day 31.12.2017 was 23,751. According to the results of our previous study [27], meningococcal vaccination uptake among Polish children 0-5 years old was 13.3%. With a confidence level (CI) of 95%, and the arbitral relative  precision of 6% points on each side, a sample of 264 parents of children 0-5 years old was needed for the purpose of this study [28]. Three hundred and fifty patients were invited to participate. Therefore, the required condition for a minimal sample size was fulfilled.

This was also incorporated to the Methods section.

4. How did the authors treat with the missing data?

Only a small number of responses were missing for most questions (1-5%). The actual number of responses for each item was included in the results, so sample sizes vary from 309 to 327. In the case of the logistic regression model - records with missing values have been omitted, in the case of the decision tree and the univariate analysis only missed values of an attribute(s) of interest have been omitted.

5. Results: Table 1: Why cut-off of 32 years old was used? If there is not any specific meaning, I would suggest keep using "mean age 32.3 years, range: 17–57 years; SD=6" mentioned in the main text.

The mean age of the study population was 32.3 years. Therefore, we arbitrary decided to choose 32 years of age as the cut off. According to the Reviewer’s comment this has been added to the Methods section of the manuscript as follows:

The mean age of the 327 participants was 32.3 years (range: 17–57 years; SD=6). Therefore, we arbitrary decided to choose 32 years of age as the cut off.

1.       Why "Owing a car" was a variable of interest in the study? Please explain.

Questions which were developed for the purpose of this study were based on a literature review, including the SAGE Working Group on Vaccine Hesitancy reporting form*. “Owning a car” was one of the variables of interest in this form; it has been also currently used by others [Domek GJ]** as a variable which may serve as a socio-economic status indicator. The majority of surveyed parents (85.3%) owned a car, and knowledge about invasive meningococcal disease in this group was significantly (p<0.001) higher compared to the group of respondents who were not car owners.

* Larson HJ et al. Measuring vaccine hesitancy: the development of a survey tool. Vaccine, 2015; 33: 4165

**Domek GJ et al. Measuring vaccine hesitancy: Field testing the WHO SAGE Working Group on Vaccine Hesitancy survey tool in Guatemala, Vaccine 2018; 36(35):5273.

7. Why stepwise selection was used? It seems the authors did not intent to build a prediction model for that. If the objective is to assess the independent association with the knowledge, all variables should include in the model with doing ANY variable selection procedure. I have no idea why a desicion tree should be presented. No additional information was drawn to the paper.

Indeed, our primary goal of using logistic regression (LR) was rather to use it to estimate variable significance, not to build a predictive model. A common LR model is sensitive regarding random (noise) variables and existing dependencies between variables. The stepwise procedure is primarily used as qualitative method for the selection of variables. The LR model can be compared with the decision tree model; the latter can be viewed as an another kind of structural selection of dependencies between variables. The results we obtained regarding the full and reduced model are very similar (see below); the same variables are significant. After introducing a selection - some of dependencies are even strengthened due to the removal of some noise attributes.

We are aware of some of restrictions of any selection procedure, but in our opinion a form of selection introduced here is required. By comparing two models (the decision tree and the LR model) it can be clearly illustrated that the model offers similar results despite the fact that dependence between variables in both models differs a little (in the decision tree we observed a conditional dependence). The decision tree structure gives some additional information. For example this tree (fig 1) shows that variables, such as "knowledge self-assessment" or "marital status" are only more significant when someone "has ever had previous information about meningococcal infection". For the abovementioned reasons we  would like to leave the decision tree model and the reduced the LR model. However, according to the Reviewer’s comment, we have added 2 more variables, so the full reduced LR model is now presented (previously in Table 4 only 4 significant variables were provided).  

---------------------------------------------------------------------------------------

Full model:

                                Estimate Std. Error z value Pr(>|z|)  

facilityTRUE                       2.19803    0.33499   2.351   0.0187 *

age                                1.02768    0.02714   1.006   0.3145  

genderTRUE                         0.52244    0.42620  -1.523   0.1277  

`marital status`TRUE               1.76620    0.37127   1.532   0.1255  

residenceTRUE                      1.74288    0.37360   1.487   0.1370  

`have a car`TRUE                   1.11140    0.51722   0.204   0.8382  

literacyTRUE                       0.29744    0.48365  -2.507   0.0122 *

employmentTRUE                     1.17942    0.36957   0.447   0.6552  

`N of children`TRUE                1.83692    0.53649   1.133   0.2570  

`Q:effective`TRUE                  0.45535    0.77457  -1.016   0.3098  

`Q:importance`TRUE                 1.62171    0.52440   0.922   0.3565  

`Q:side effects`TRUE               0.98774    0.31589  -0.039   0.9688  

`Q:ever heard`TRUE                 2.84645    0.40842   2.561   0.0104 *

`Q:knowledge self-assessment`TRUE  2.59086    0.46714   2.038   0.0416 *

sourcesTRUE                        1.63851    0.35526   1.390   0.1645  

facilityTRUE                      1.157355347 4.3262703

age                               0.974735166 1.0850880

genderTRUE                        0.224965101 1.2082629

`marital status`TRUE              0.850647837 3.6691397

residenceTRUE                     0.836947018 3.6431871

`have a car`TRUE                  0.397868942 3.0608545

literacyTRUE                      0.113260033 0.7623638

employmentTRUE                    0.564905005 2.4213911

`N of children`TRUE               0.638073705 5.3043806

`Q:effective`TRUE                 0.090406295 2.0079634

`Q:importance`TRUE                0.574296303 4.5950503

`Q:side effects`TRUE              0.531450131 1.8402110

`Q:previous info`TRUE                1.291437529 6.4431681

`Q:knowledge self-assessment`TRUE 1.075956301 6.8266599

sourcesTRUE                       0.818571745 3.3115951

---------------------------------------------------------------------------------------

Reduced model:

Coefficients:

                                Estimate Std. Error z value Pr(>|z|)    

facilityTRUE                        2.0629     0.3226   2.245 0.024794 *  

marital statusTRUE                1.7177     0.3183   1.699 0.089226 .  

residenceTRUE                       1.7039     0.3470   1.536 0.124554    

literacyTRUE                        0.2511     0.4073  -3.392 0.000693 ***

`Q:previous info`TRUE                  3.8263     0.3540   3.790 0.000150 ***

`Q:knowledge self-assessment`TRUE   2.8139     0.4331   2.389 0.016916 *  

---

                                     2.5 %    97.5 %

(Intercept)                       0.07463234 0.5636794

facilityTRUE                      1.11082429 3.9526179

marital status TRUE              0.91723117 3.2084854

residenceTRUE                     0.86207852 3.3776717

literacyTRUE                      0.11020731 0.5492912

`Q:previous info`TRUE                1.93511886 7.8004209

`Q:knowledge self-assessment`TRUE 1.25065110 6.9376699

Round  2

Reviewer 2 Report

Thank you for addressing the majority of my concerns and comments.

I have a few remaining minor comments at this stage:

Various ways to increase response rates are documented in the literature

Refer to 'Mann Whitney's test' as the Mann-Whitney U test

I think you should say the questionnaires were 'completed' by the researchers rather than 'self-completed' by the researchers ('self-completed' tends to mean they were completed by the respondents/participants themselves). Readers should also appreciate how consistency of approach was achieved during data collection (several different researchers were helping people complete it/training for the researchers beforehand about the level of help that could be provided).

Make sure it is very clear for the reader to see the number of people who were invited to participate

Author Response

We would like to thank the Reviewer for his valuable comments which helped us to improve the manuscript.

Please, find below our point-by-point response to the reviewer’s comments.

Response 1:

Various ways to increase response rates are documented in the literature.

According to the literature review, the odds of response are increased through using monetary and non-monetary incentives, shorter questionnaires, mentioning an obligation to respond and university sponsorship, assurance of confidentiality, using textual representation of response categories [Edwards]. Contrary, the odds of response are reduced when the questionnaire includes questions of a sensitive nature.

Edwards PJ, Roberts I, Clarke MJ, DiGuiseppi C, Wentz R, Kwan I, Cooper R, Felix LM, Pratap S. Methods to increase response to postal and electronic questionnaires. Cochrane

Database Syst Rev 2009;3:MR000008.

Edwards P. Questionnaires in clinical trials: guidelines for optimal design and administration. Trials 2010, 11:2 http://www.trialsjournal.com/content/11/1/2

According to the Reviewer’s comment this has been added to the Methods section of the manuscript as follows:

A structured anonymous self-administered questionnaire was used as the main data collection instrument. It was designed by the authors with the use of literature review [2,29]. To increase response rate [30,31] parents were assured of confidentiality of the survey. In addition, questions of a sensitive nature were omitted. Questionnaires (completed by respondents) were administered to the parents by three researchers (M.D-D., K.T. and A.S.), present at a dedicated PCC at the time the survey was conducted. In case of any queries, parents were provided with relevant explanation, as well as any further information that would allow them to correctly understand and properly answer the survey questions. The researchers were trained beforehand about the level of help that could be provided regarding the fulfillment of study questionnaires.

Response 2:

Refer to 'Mann Whitney's test' as the Mann-Whitney U test.

According to the Reviewer’s comment this has been changed in the Abstract/the manuscript.

Response 3:

I think you should say the questionnaires were 'completed' by the researchers rather than 'self-completed' by the researchers ('self-completed' tends to mean they were completed by the respondents/participants themselves).

We would like to apologize for the syntax error in this sentence. According to the Reviewer’s comment this has been changed in the Instrument and Data Collection section of the manuscript as follows:

A structured anonymous self-administered questionnaire was used as the main data collection instrument. (…) Questionnaires (completed by respondents) were administered to the parents by three researchers (M.D-D., K.T. and A.S.), present at a dedicated PCC at the time the survey was conducted.

Response 4:

Readers should also appreciate how consistency of approach was achieved during data collection (several different researchers were helping people complete it/training for the researchers beforehand about the level of help that could be provided).

According to the Reviewer’s comment this has been changed in the Instrument and Data Collection section of the manuscript as follows:

The researchers were trained beforehand about the level of help that could be provided regarding the fulfillment of study questionnaires.

Response 5:

Make sure it is very clear for the reader to see the number of people who were invited to participate.

According to the Reviewer’s comment this has been changed in the Abstract:

Initially 350 parents were invited, 327 agreed to participate in the study (the response rate 93.4%).

Reviewer 3 Report

The authors addressed well to my comments except the last one. If using the logistic regression is to determine the significance, no variable selection should be used as the procedure may drop the variables that confounding the true effects, especially it is not a randomized trial. It is obviously not the problem of the robustness of using which variable selection methods. I recommend a statistical review for the approach and results.

Author Response

We would like to thank the Reviewer for all valuable comments which helped us to improve the manuscript.

Please, find below a point-by-point response to the reviewer’s comments.

The authors addressed well to my comments except the last one. If using  the logistic regression is to determine the significance, no variable  selection should be used as the procedure may drop the variables that  confounding the true effects, especially it is not a randomized trial.  It is obviously not the problem of the robustness of using which  variable selection methods. I recommend a statistical review for the  approach and results.

According to the Reviewer’s comment the full model was used  (no variable  selection) as follows:

Table 4. Logistic regression model: association of the knowledge level about meningococci with selected variables (odds ratio (OR) estimates* and 95% confidence intervals (CIs) of OR estimates); Poland, 2018; n = 327.

Variable

OR

                     CI

Parental age: >32

1.03

                   0.97-1.09

Gender: father

0.52                    

                   0.22-1.22

Marital status: married/cohabitating

1.77

                   0.85-3.67

Residence: urban

1.74  

                   0.84-3.64

Education: high school/university

3.37

                 1.31–8.85

Employment: yes

1.18

                   0.56-2.42

Owning a car: yes

1.11

                   0.40-3.06

Number of children: ≤2

1.84

                   0.64-5.30

Facility: urban

2.20

                   1.16-4.33

Had ever received information about IMD

2.85

                   1.29-6.44

Self-assessed knowledge: good

Source of knowledge: GP

Concerned about the safety of MV: yes   

2.59

1.64

0.99                   

                   1.08–6.83

                   0.82-3.31

                   0.53-1.84

* Odds ratio (OR) = ratio between the two categories tested in each variable, controlling for other variable.

Relevant changes have been made in the Abtract, as well as in the Results section. We also deleted the fragment “a stepwise backward selection procedure was performed” from the description of the statistical methods.

Furthermore, according to the Reviewer’s comment from the first round of the revision process (I have no idea why a decision tree should be presented. No additional information was drawn to the paper)  the decision tree was deleted from the manuscript.